# Properties and Printability of the Synthesized Hydrogel Based on GelMA

**DOI:** 10.3390/ijms24032121

**Published:** 2023-01-20

**Authors:** Nadezhda V. Arguchinskaya, Elena V. Isaeva, Anastas A. Kisel, Evgeny E. Beketov, Tatiana S. Lagoda, Denis S. Baranovskii, Nina D. Yakovleva, Grigory A. Demyashkin, Liudmila N. Komarova, Svetlana O. Astakhina, Nikolai E. Shubin, Peter V. Shegay, Sergey A. Ivanov, Andrey D. Kaprin

**Affiliations:** 1A. Tsyb Medical Radiological Research Center–Branch of the National Medical Research Radiological Center of the Ministry of Health of the Russian Federation, 249036 Obnisk, Russia; 2Obninsk Institute for Nuclear Power Engineering, National Research Nuclear University MEPhI, 249039 Obninsk, Russia; 3National Medical Research Radiological Center, Peoples’ Friendship University of Russia (RUDN University), 117198 Moscow, Russia; 4Federal State Autonomous Educational Institution of Higher Education I.M. Sechenov, First Moscow State Medical University of the Ministry of Health of the Russian Federation, 119991 Moscow, Russia; 5System Products for Construction, 249020 Vorsino, Russia; 6National Medical Research, Radiological Centre of the Ministry of Health of the Russian Federation, 249036 Obninsk, Russia

**Keywords:** hydrogel, GelMA, extrusion 3D bioprinting, cytocompatibility, biocompatibility

## Abstract

Gelatin methacryloyl (GelMA) has recently attracted increasing attention. Unlike other hydrogels, it allows for the adjustment of the mechanical properties using such factors as degree of functionalization, concentration, and photocrosslinking parameters. In this study, GelMA with a high degree of substitution (82.75 ± 7.09%) was synthesized, and its suitability for extrusion printing, cytocompatibility, and biocompatibility was studied. Satisfactory printing quality was demonstrated with the 15% concentration hydrogel. The high degree of functionalization led to a decrease in the ability of human adipose-derived stem cells (ADSCs) to adhere to the GelMA surface. During the first 3 days after sowing, proliferation was observed. Degradation in animals after subcutaneous implantation was slowed down.

## 1. Introduction

Tissue engineering and regenerative medicine aim to repair damage to organs and tissues caused by a disease or an injury. Additive technologies are widely used to create tissue-engineering structures. The 3D structure is generated layer by layer on the basis of a computer-aided design model (CAD). This technology includes various strategies: laser, inkjet, and extrusion printing, stereolithography, and electrospinning [1]. In order to achieve the desired result, various hydrogels are used, which have the ability to provide mechanical support to cells and mimic the native extracellular matrix [2,3]. The materials used to produce hydrogels may be of natural or synthetic origin. Natural polymers include materials based on protein (gelatin, collagen, fibrin, and silk fibroin) or polysaccharides (hyaluronic acid, chondroitin sulfate, alginate, and chitosan) [4,5,6]. Synthetic hydrogels are obtained from polyethylene glycol, polyvinyl alcohol, and polyacrylamide and its derivatives, as well as pluronics [7,8,9,10].

A commonly used natural material is gelatin, which is a mixture of proteins obtained by acid or alkaline hydrolysis of collagen [11]. The advantages of gelatin are its availability, low cost, biodegradability, biocompatibility, and low immunogenicity. However, it is not stable at +37 °C due to its low melting point of 31.7–34.2 °C. Gelatin hydrogel has poor mechanical strength, which makes it difficult to use at physiological temperatures [12].

In recent years, much research has focused on the gelatin derivative, gelatin methacryloyl (GelMA). GelMA, like gelatin, has bioactive sequences (e.g., arginine–glycine–aspartic acid) for cell attachment and matrix metalloproteinase-responsive sites responsible for cell-mediated degradation [13,14]. Under UV irradiation in the presence of a photoinitiator, the hydrogel acquires the property of forming a 3D structure. The advantage of GelMA compared with other hydrogels is the ability to control mechanical properties by changing the concentration [14,15], degree of functionalization (DoF) [15,16], type of photoinitiator used (Irgacure 2959, lithium acylphosphinate, or riboflavin) [17,18,19,20], and UV-exposure time [16].

The GelMA synthesis was first described by A.Van Den Bulcke et al. [21]. The other methods are its modifications and just complement the original. GelMA is prepared by reacting gelatin with methacrylic anhydride in an alkaline or neutral buffer solution. This can be phosphate-buffered saline (PBS) [22] or carbonate–bicarbonate buffer [23]. During the synthesis, a by-product, methacrylic acid, is formed, which lowers the pH of the solution [23]. It is necessary to take into account and maintain the proper pH value. Residual methacrylic anhydride and methacrylic acids are removed by dialysis. Further, a foam-like material is obtained in a freeze-drying process. The hydrogel polymerizes quite quickly under the action of UV in the presence of a photoinitiator and remains stable at physiological temperatures [24]. By changing the amount of methacrylic anhydride during synthesis, it is possible to obtain gelatin with different DoFs. An increase in the percentage of methacryloyl substitution from 20 to 80 leads to the formation of more rigid and durable hydrogels [17,25]. The biomaterial can be further adapted to form specific tissues by creating desired physicochemical properties [13,26].

The aim of this work was a comprehensive study of the synthesized GelMA, including the degree of functionalization, viscosity, printability, cytocompatibility, and biocompatibility. We consider the study performed with the use of animals to be an important part since in most similar works, the authors limit themselves to studying only the cytocompatibility of hydrogels.

## 2. Results

### 2.1. Assessment of GelMA DoF

Colorimetric analysis of ninhydrin can detect protein functionalization by the loss of free amines. Primary amines react with a small ninhydrin molecule to form a purple dye. Using the calibration curve, the concentration value was determined from the value of the optical density of the hydrogel sample. Next, the proportion of free amines remaining after conjugation and DoF were calculated. Figure 1A shows an example of staining for gelatin at a concentration of 1–10 mg/mL. Figure 1B shows optical absorption as a function of gelatin concentration. The apparent concentration of the gels was found from the optical density of GelMA samples with a nominal concentration of 10 mg/mL (Figure 1C). 

The DoF percentages for the synthesized and commercial gels were 82.75 ± 7.09 and 79.44 ± 3.56 (the manufacturer declared a substitution rate of 80%), respectively.

### 2.2. Assessment of Rheological Properties

#### Viscosity

Viscosity is one of the fundamental properties of hydrogels used for bioprinting. Figure 2 shows that the viscosity of gelatin depends on temperature and concentration. As the concentration increased and the temperature decreased, the viscosity increased continuously. For concentrations of 7.5–15%, the viscosity increased sharply at temperatures below +32 °C. The viscosity of the synthesized GelMA did not grow so rapidly and remained low at a temperature of +21 ℃, which indicates a change in the rheological properties of the hydrogel caused by gelatin functionalization. 

### 2.3. Printability

Since the rheological properties of the synthesized GelMA differed significantly from those obtained for the original gelatin, it was necessary to verify its printability for other conditions. The most common printing temperature for hydrogels, 0 ℃, was chosen as the printing temperature. The extrusion was performed through a 21G needle. It was necessary to make sure that the hydrogel formed a filament, not droplets, which is the key requirement for extrusion bioprinting [27]. Figure 3A shows the formation of a filament by the synthesized GelMA hydrogel during extrusion. 

Thus, the hydrogel has a viscosity sufficient for printing, and our assumption about the need to lower the temperature at which printing should be performed is correct.

#### 2.3.1. Maximum Printing Speed of the First Layer

The print speed should ensure a continuous supply of the filament. Broken lines and print distortion indicate that the selected speed is not applicable. Printing lines of a given height in one layer at different speeds (from 5 to 20 mm/s, with a step of 2.5 mm/s) is shown in (Figure 3B). Figure 3C shows that printing was reproduced over the entire range of speeds examined.

#### 2.3.2. Object Geometry after Incubation

Figure 3D, Figure 3E show the appearance of objects printed with 15% GelMA before and after 72 h incubation. Table 1 presents the results of measuring the area of objects at the same time. The data obtained (Table 1) show the presence of a slight increase in the area after incubation, which was not statistically significant. At the same time, it should be noted that the area of products after printing differed from the set value (0.640 cm^2^ vs. 0.721 cm^2^) by 12.7%, and after incubation by 28.1%. Thus, the shape of the printed objects was preserved visually, while the differences from the parameters specified in the software of the printer after printing could be associated with the ongoing polymerization of the hydrogel. It is also quite obvious that hydration (swelling) of the hydrogel additionally occurs during incubation. This is also characteristic of other hydrogels, such as pure collagen, which polymerizes due to a pH shift [4].

#### 2.3.3. Printing at an Angle

Figure 3F shows the test panel and the panels obtained during the printing of Figure 3G, and Figure 3H shows lines with sharp, right, and obtuse angles. It can be seen that when printing at a sharp angle at a speed of 5 mm/s, the print quality deteriorated. Some areas overlapped (the overlay of the hydrogel in Figure 3G), which, in turn, resulted in an uneven layer height. Due to the fact that the layer height error will accumulate, it will lead to distortion of the given geometric shape of the object.

However, when the printing speed was reduced to 1 mm/s, its quality at an acute angle improved, and no significant hydrogel overlap was observed (Figure 3H). This may be due in part to the longer total UV exposure time during printing at 1 mm/s, which may have contributed to better curing of the hydrogel.

### 2.4. Assessment of Cytocompatibility

Data on the number of ADSCs on Days 3 and 7 of cultivation in all four groups are shown in Figure 4. These results show that at 3 days, the number of cells in the three groups decreased relative to the initial number and was 93%, 54%, and 59% in the group with unmodified gelatin, with synthesized GelMA, and with commercial GelMA, respectively. This shows that not all cells attached to the surface of the hydrogels and retained their viability during the first 3 days of cultivation. By the 7th day, the number of cells relative to the 3rd day increased by 1.79 ± 0.16, 1.86 ± 0.27, 1.64 ± 0.24, and 1.21 ± 0.25 times in groups without gelatin, with gelatin, GelMA (synthesis), and GelMA (Sigma), respectively (Figure 4). Pairwise comparison of the groups using Tukey’s criterion showed no significant differences in the number of cells between the groups with GelMA synthesis and GelMA (Sigma) at Day 3 and Day 7. However, the number of cells in the group that were plated on unmodified gelatin was significantly higher than in the groups with GelMA. The number of cells in the group without gelatin was also significantly higher compared with the group with unmodified gelatin. Thus, the addition of unmodified gelatin was not indifferent to the cells. Functionalization of gelatin with methacrylic anhydride led to a decrease in the ability of ADSCs to adhere and proliferate during the first 3 days after seeding.

### 2.5. Biocompatibility Assessment In Vivo

Scaffolds with surrounding tissues were removed on Days 10, 17, and 33 after implantation. The results of histological studies are shown below. 

On the 10th day, a thin connective tissue capsule was formed around the implant located in the subcutaneous adipose tissue of rats (Figure 5A). The capsule contains a large number of microcirculation vessels, fibrous structures, and fibroblasts. Scaffold fragments of different sizes are detected, surrounded by multinucleated cells of resorption of foreign bodies (Figure 5B). Among the elements of the fibroblastic series, the presence of a small amount of round cell leukocyte infiltrate is observed.

On the 17th day after implantation, scaffold degradation was not observed (Figure 6A). Rather large fragments of the implant remain in the connective tissue capsule, and the number of multinucleated macrophages that destroy these fragments increases (Figure 6B). The presence of a small amount of round cell infiltrate mast cells in the perivascular spaces of large vessels is noted.

On the 33rd day after implantation, the scaffold is surrounded by the connective tissue capsule, which in its immediate vicinity, consists of a small number of oxyphilic, more mature collagen fibers, with fibrocytes rarely located between them (Figure 7A–C). In the capsule, fragments of the implant can be observed (Figure 7D), as in the previous periods of the study. Next to them are visible multinucleated macrophages that destroy the scaffold (Figure 7E), the number of which visually seems to be reduced, and a round cell infiltrate (Figure 7F).

Thus, the results of the histological study allow us to conclude that the scaffold material is slowly degrading and does not cause the development of a pronounced inflammatory reaction. After 33 days, a narrow layer of mature connective tissue surrounding the implant is formed in the connective tissue capsule, in which the processes of its degradation with the help of multinucleated cells of foreign body resorption are significantly slowed down.

## 3. Discussion

The addition of methacrylate groups to the amine-containing side groups of gelatin is used to transform it during polymerization into a hydrogel that is stable at +37 °C [25]. Therefore, the necessary procedure after the synthesis of GelMA is to determine the degree of methacrylation or functionalization, since it affects the polymerization rate, rigidity, and mechanical properties of the hydrogel [16,22]. DoF strongly depends on the reaction conditions, the source of gelatin (pork, bovine, or fish), and on the amount of methacrylic anhydride used for synthesis [15,18,28]. As a rule, GelMA with a degree of substitution of 20–80% is used to create stable hydrogels [17,25]. With an increase in the degree of methacrylation within these limits, the rigidity of GelMA hydrogels increases and the swelling coefficient decreases [25]. Y. Chen et al. [16] showed that with an increase in the degree of functionalization, the elastic modulus of GelMA hydrogels under compression increased (49.8%–2.0 ± 0.18 kPa, 63.8%–3.2 ± 0.18 kPa, and 73.2%–4.5 ± 0.33 kPa). In the work of I. Pepelanova et al. [15], a decrease in the degradation time of the GelMA hydrogel with an increase in the degree of functionalization was described. The authors also note that the degree of methacrylation of GelMA must be taken into account in bioprinting since bioinks from GelMA with a low degree of functionalization showed low viscosity, which is not suitable for the bioprinting process. An increase in the degree of methacrylation led to an improvement in the viscosity of the bioink, which, in turn, improved such a property of the hydrogel as shear thinning [15]. This characteristic is important for maintaining cell viability during printing. Since we intended to use the GelMA synthesized by us for bioink production and printing without addition of other bioink components responding for printability, this remark was important for us. The degree of functionalization of the GelMA obtained by us was 82.75 ± 7.09%, which should have provided the necessary viscosity of the hydrogel. 

The rheological properties of hydrogels and, first of all, viscosity are one of the most important parameters. Viscosity determines the suitability of the gel for extrusion bioprinting and its quality [26,29]. The viscosity of the hydrogel at the moment of printing is of key importance here. In addition to DoF, it depends on the temperature at which printing takes place and the concentration of the hydrogel. According to the data given in C. Chang et al. [30], the viscosity range of hydrogels suitable for extrusion printing is very wide, from 30.0 to > 6.0 × 10^7^ mPa × s [30]. In our study, the viscosity of 15% hydrogel from synthesized GelMA was 22.5 mPa × s at 21 ℃, so we decided to lower the temperature at which printing should take place. T. Billiet [31] describes one approach to improve the printability of GelMA hydrogels on the basis of the temperature-dependent sol–gel transition inherent in gelatin. The authors note that cooling bottom layers of the scaffold to 5 °C immediately after deposition enhanced the physical crosslinking of GelMA and provided sufficient mechanical strength for the printed structures. This method was suitable for GelMA concentrations from 10% to 20%. Without cooling, GelMA structures of 10% and 15% concentrations completely collapsed due to insufficient mechanical integrity. It should be added that the degree of gelatin methacrylation in this study was lower at 62%. N. Rajabi et al. [26], in their review, report that lowering the temperature to 0° increases self-healing, shear thinning, and GelMA printing accuracy. In the studies of W. Schuurman et al. [17], the low viscosity GelMA solution formed droplets at the tip of the needle, causing the material to spread over the printed surface, and printing could not be accomplished. The authors draw attention to the possibility of using the thermally sensitive properties of GelMA during bioprinting to provide mechanical support until covalent photocrosslinking is complete. It is noted that photocrosslinking of polymer chains of GelMA, which is in a thermal gel state, retains the triple helical conformation and their biological significance. B. Lee et al. [32] report that the printability of GelMA is significantly sensitive to temperature changes. We investigated the printability of 15% GelMA hydrogel at the following temperatures: +4 ℃—dispenser and 0 ℃—table. Under these conditions, the hydrogel was extruded in the form of a continuous filament in the entire investigated range of speeds. However, the print quality at an angle was higher at a lower speed. 

GelMA’s properties and printability are significantly affected by its concentration. As the concentration increases, the printing accuracy and the ability of the printed structures to retain the specified geometric shape increases. For bioprinting, GelMA is most often used in concentrations of 10–30% [14,33,34]. The printability experiments conducted by L. Bertassoni et al. [35] show that printing can be performed by GelMA hydrogels with concentrations of 7 to 15%. Lower concentrations did not promote the formation of homogeneous filaments. Another work also noted the poor suitability of 5% GelMA for bioprinting because of its low viscosity [15]. As mentioned above, 15% GelMA in our work showed its printability. The printed objects retained the specified geometric shape after 72 h incubation in nutrient medium at +37 °C. The swelling of the hydrogel was insignificant, which is consistent with the data of J. Nichol et al. [25] on the dependence of this indicator on DoF.

Cytocompatibility and biocompatibility are important indicators of the suitability of biomaterials for creating tissue-engineered constructs based on them. Studies using different cell lines, such as chondrocytes [18,36], keratinocytes [37], human umbilical vein endothelial cells (HUVEC) [25,38,39], fibroblasts [40], myoblasts [41], and mesenchymal stem cells (MSCs) [42] showed high cell viability (>80%), encapsulated into GelMA hydrogels or cultured on their surface [18,25,28,37,39]. The ability to bind to framework materials is essential for the survival and function of cells in artificial tissues [43]. However, this ability also depends on the concentration of the hydrogel and its stiffness. Y. Wu et al. [44] showed that the adhesion ability of PC12 nerve cells decreased with increasing GelMA hydrogel concentration from 5 to 30%.

We chose adipose-derived MSCs because of their ability to differentiate into several cell types (e.g., osteoblasts, chondroblasts, and adipocytes) [45], as well as to study the compatibility of GelMA with human cells. At the same time, concomitant side effects should be considered in the clinical use of MSCs [46]. In our study, these cells demonstrated low adhesion to the surface of GelMA hydrogels and proliferative activity during the first 3 days of cultivation. This may be due to several factors, including the properties of the gelatin itself and those that it acquires during the functionalization and photocrosslinking processes. This was equally true for both the synthesized and commercial versions of the biomaterial.

The properties of unmodified gelatin used for synthesis differ depending on the method of collagen hydrolysis by which it is obtained: acidic (type A) or alkaline (type B). Gelatins A and B promote the binding of various types of growth factors, as well as the proliferation of various types of cells [47]. Different concentrations of gelatin, DoF, type of photoinitiator, and irradiation time affect the rigidity of hydrogels [13], changing the quantity of bioactive sequences for cell binding [25]. One advantage of GelMA is the distribution of cell adhesion sites throughout the hydrogel on all polymer chains, potentially increasing the likelihood of cell binding. However, cell spreading and migration will be limited by their ability to degrade and remodel the matrix, which varies greatly among different cell types [25]. Finally, cell viability is affected by the type of photoinitiator and its concentration. Much of the published literature, as well as our work, uses GelMA with the Irgacure 2959 photoinitiator. It is believed to have relatively low cytotoxicity compared with other photoinitiators [48]. At the same time, the long-term effect of Irgacure 2959 on cells has not yet been fully studied [13].

The results of in vivo biocompatibility experiments are consistent with those obtained in vitro. Scaffolds made of 10% synthesized GelMA obtained by the molding method and implanted under the skin in rats were very slowly resorbed without causing a pronounced inflammatory reaction. Thirty-three days after implantation, large fragments of scaffolds surrounded by a connective tissue capsule were still preserved. This can probably be explained by the high degree of functionalization of gelatin. It is known that as the degree of metacrylation increases, the biodegradation time of GelMA structures is prolonged [49,50]. The second probable reason could be the absence of encapsulated cells secreting matrix metalloproteinases to which GelMA hydrogels are sensitive [49]. In our other study (data not published), GelMA scaffolds with rat chondrocytes encapsulated in them completely dissolved within 26 days of subcutaneous implantation in rats. In some studies, a slight inflammatory reaction was observed after implantation of the scaffolds in animals [39,51]. S. Heltmann-Meyer et al. note the presence of macrophages with both pro-inflammatory and anti-inflammatory phenotypes without a specific pattern of distribution or predominance of one subtype [51]. An extensive in vitro study showed good biocompatibility of GelMA based on gelatin type B, while GelMA based on gelatin type A caused inflammatory reactions [52]. At the same time, the natural inflammatory response may be useful for recruiting endogenous cells for tissue regeneration [53]. Many studies report successful tissue formation within a few weeks after implantation of GelMA-based scaffolds in animals. It provided the formation of nervous tissues [54], vasculature [38,55], skin [37], bones [19,33], and cartilage [6,36,56].

## 4. Materials and Methods

### 4.1. Synthesis of GelMA

Synthesis of GelMA was carried out on the basis of the methodology described in the article by D. Loessner et al. [22]. Pig skin gelatin powder, Bloom strength ~300 g, type A (Sigma-Aldrich, G1890, St. Louis, MO, USA) was used for synthesis. Gelatin at a final concentration of 10% was soaked in PBS (pH 7.4, ECOservice, Murmansk, Russia) and left at room temperature for 60 min to swell. Then, it was stirred on a magnetic stirrer with heating to +50 °C for 40–60 min until complete dissolution. Then, while controlling the pH, methacrylic anhydride (Sigma-Aldrich, 276685, St. Louis, MO, USA) was added dropwise with vigorous stirring—580 μL per 1 g of gelatin. The pH of the solution was maintained at 7.4 by adding 1M NaOH. The solution was allowed to react for 1 h with vigorous stirring at +50 °C. After that, it was transferred to 15 mL centrifuge beakers (Corning, New York, NY, USA) and unreacted methacrylic acid was removed by centrifugation at 2500× *g* for 3 min. Methacrylic acid precipitated as a white viscous precipitate. The supernatant was placed in a beaker and diluted with distilled water preheated to +40 ℃. The solution was transferred to a 12,000–14,000 kDa MWCO dialysis bag (SERVAPOR, Heidelberg, Germany) and subjected to dialysis in distilled water at room temperature for 7 days to ensure complete removal of impurities. The water was changed once per day. The solution was then collected, and the pH was adjusted to 7.4. Next, the solution was sterilized using filter nozzles on a polyethersulfone (PES) syringe with a pore size of 0.2 μm (Thermo FS, San Diego, CA, USA). The sterile solution was divided into aliquots, frozen at −80 ℃, and then transferred to a 15-SRC-X lyophilic dryer (VirTis, Gardiner, NY, USA), followed by drying in a vacuum chamber for 48 h in a “floating” mode with a temperature change from −40° to +20 °C. The lyophilized samples were stored in a dark container at −20 ℃. In order to avoid light, all procedures were carried out in a darkened room, and the dishes were wrapped in foil.

### 4.2. Assessment of GelMA DoF

In order to assess the degree of functionalization (DoF) of GelMA, a colorimetric analysis of ninhydrin was used according to the method described by J. Zatorski et al. [57]. Our synthesized GelMA was compared with a commercial variant of GelMA (Sigma-Aldrich, 900496, St. Louis, MO, USA). The control was the original (unmodified) gelatin. Unmodified gelatin, synthesized, and commercial GelMA were dissolved at a concentration of 10 mg/mL in PBS. In order to obtain a standard (calibration) curve, unmodified gelatin was diluted in PBS at a concentration of 0 to 10 mg/mL in increments of 1 mg/mL. PBS was without calcium and magnesium. GelMA samples were not diluted. A solution of ninhydrin (U485472.0025, Dia-m, Moscow, Russia) 20 mg/mL in 70% ethanol (ZAO RFK, Moscow, Russia) was added to the samples in a ratio of 1:8. The final concentration of ninhydrin was 2.2 mg/mL. The samples were incubated in a thermostat (Termite, DNA technology, Russia) at +70 °C for 20–30 min until a purple color appeared. Then, they were transferred to 96-well round bottom plates (Corning, New York, NY, USA), and the optical density was measured at 570 nm using a Multiskan FC microplate reader (Thermo Scientific, San Diego, CA, USA). The measurements were carried out in three repetitions.

A standard gelatin dilution curve was built to calculate the degree of functionalization. The percentage of DoF was determined by Formula (1).
(1)$$DoF\left(\%\right)=100\cdot \left(1-\frac{Apparent\;Sample\;Conc.}{Nominal\;Sample\;Conc.}\right)\;$$
where the apparent concentration was determined by the corresponding optical density value on the gelatin standard curve, and the nominal concentration is the concentration at which the solution was prepared.

### 4.3. GelMA Hydrogel Preparation

In order to prepare the hydrogel, the photoinitiator Irgacure 2959 (Sigma-Aldrich, 410896, St. Louis, MO, USA) was pre-dissolved at a concentration of 0.1% in hot distilled water (+70 ℃) for 15 min in a solid-state thermostat (Termite, DNA Technology, Moscow, Russia), periodically stirring for vortex (Biosan, Riga, Latvia), after which they were cooled at room temperature in a place protected from light. After cooling, the solution was added to a sample of lyophilized GelMA and left overnight at +4 ℃ to swell. The next day, the solution was kept at +37 ℃ for 3 h in a solid-state thermostat (Termite, DNA Technology, Moscow, Russia) to dissolve GelMA, after which it was cooled in a dispenser of a Rokit Invivo 3D printer (Seoul, Republic of Korea) and used for printing or molding.

### 4.4. Assessment of Rheological Properties: Viscosity

In order to determine the viscosity, weighed portions of unmodified gelatin and synthesized GelMA were dissolved in a water bath at +50 °C. Gelatin hydrogels were prepared at a concentration of 5, 7.5, 10, 12.5, and 15% and GelMA–15%. After that, they were placed in the measuring unit of a Brookfield DV-II+PRO viscometer (Middleboro, MA, USA), and the viscosity was determined. The measurement was carried out for cooling in the temperature range of 48.5–21.5 ℃.

### 4.5. Printability Rating

The suitability of a hydrogel synthesized at 15% strength GelMA for extrusion printing was investigated. Printing was carried out on a Rokit Invivo 3D printer (Seoul, South Korea) with software version 1.68. The g-code was created using the NewCreatorK program version 1.57.63. The temperature of the printing table was 0 ℃, the dispenser was +4 ℃, the printer chamber was room temperature. The standard printing speed was 5 mm/s, and the needle diameter was 21G (inner diameter 514 µm). The objects were printed with a syringe dispenser onto glass or into a Petri dish (Corning, New York, NY, USA) with a diameter of 6 cm. Before printing the main object, a contour stroke (“Skirt”) was printed, which made it possible to prepare the extruder and achieve smooth extrusion of the filament onto the surface. During printing, the products were subjected to UV irradiation (Photo Curing with Light Emitting Diode, Wavelength: 365 nm, ROKIT HEALTHCARE, Seoul, South Korea). At the end of printing, all products were photographed.

#### 4.5.1. Maximum Print Speed of the First Layer

In order to investigate the limitations of the printing speed of the first layer, 6 cm long lines were printed in two rows. The range of 5–20 mm/s with a step of 2.5 mm/s was used as the estimated speed values: the value of 5 mm/s is a generally accepted standard, and the value of 20 mm/s is a technological limitation for this printer model. Print quality was assessed visually.

#### 4.5.2. Preservation of the Geometric Parameters of the Product after Incubation

The ability of the material to maintain the geometric parameters specified during printing after a 72 h incubation at +37 °C was evaluated by changing the area of the object. For this purpose, an article 8 × 8 × 0.2 mm in size was printed into a Petri dish (Corning, New York, NY, USA) with a diameter of 6 cm. Immediately after printing was completed, the dish with the article was filled with DMEM medium (PanEco, Moscow, Russia) and placed in a CO_2_ incubator (Sanyo, Tokio, Japan). The object area was assessed before and after incubation using ImageJ version 1.52a.

#### 4.5.3. Printing at an Angle

To assess the quality of printing at an angle, a model was developed that allows simultaneous printing at angles of 30, 60, 90, 120, and 150 degrees. Printing was evaluated with print speeds of 1 and 5 mm/s.

### 4.6. Culture of ADSCs

Experimental studies to assess cytocompatibility were performed using ADSCs obtained from the National Research Center for Rehabilitation and Balneology of the Russian Ministry of Health. The procedure for cell isolation and cultivation, as well as the study of the phenotypic profile, are described in [4]. Briefly, cells were cultured in DMEM medium (PanEco, Moscow, Russia) with a glucose content of 1 g/L supplemented with 10% fetal bovine serum (Biosera, Nuaille, France), penicillin–streptomycin (100 U/mcg/mL), and glutamine (150 μg/mL) in a CO_2_ incubator at +37 °C according to the standard method. ADSCs of the 6th passage were used for the experiment.

### 4.7. Assessment of Cytocompatibility

The cytocompatibility of the synthesized and commercial GelMA was studied in comparison with unmodified gelatin and DMEM medium (PanEco, Moscow, Russia) in terms of such indicators as the ability to adhere and the proliferative activity of ADSCs. Hydrogels, 10% concentration, were prepared in accordance with the described method and were poured into the wells of 24-well culture plates (Nunc, Roskilde, Denmark) at 350 µL per well. Synthesized and commercial GelMA was irradiated with UV (365 nm) for 10 min in the chamber of a Rokit Invivo 3D printer. The temperature of the 3D printer stage on which the tablet was placed was 0 ℃. The unmodified gelatin was not irradiated with UV but kept on the printer stage at the same temperature until solidified. After polymerization, the height of the hydrogel layer in each well, 1.5 cm in diameter, was 0.2 cm. ADSCs were seeded on top of hydrogels and in a plate without gelatin in the amount of 2.2 × 10^4^ cells per well. The cells were incubated in a 1 g/L glucose content DMEM medium (PanEco, Moscow, Russia) supplemented with 10% fetal bovine serum (Biosera, Nuaille, France). After 3 and 7 days, the medium was removed, and the cells were removed with a trypsin-EDTA solution (PanEco, Moscow, Russia) and counted.

### 4.8. Biocompatibility Assessment In Vivo

#### 4.8.1. FDM Printing

The day before the experiment, forms were printed from PLA (polylactide, filament diameter—1.75 mm) by the FDM (Fused Deposition Modelling) method, which were hollow cylinders of 5 mm height with a solid base of 1 mm thickness, inner diameter 8.5 mm, and wall thickness 0.4 mm (Figure 8A). 

Printing was carried out on a Rokit Invivo 3D bioprinter. The model was sliced using the NewCreatorK program version 1.57.63. Printing was performed on a flat printing table without heating, and the temperature of the extruder was +210 °C. The first layer was printed at a speed of 5 mm/s, then the object was printed at 10 mm/s. The type of filling is concentric. The thickness of one printed layer is 0.2 mm with the same nozzle diameter. Before printing the object, the Skirt program was executed to minimize the defect during the application of the first layer. In order to avoid sample contamination, the interior of the printer was sterilized before printing using a built-in UV lamp with a wavelength of 254 nm. The finished forms were sterilized by ultraviolet light with the same wavelength for 2 h.

#### 4.8.2. Scaffold Preparation and Implantation in Animals

For implantation in animals, scaffolds were prepared by molding. The hydrogel of the synthesized GelMA (10%) was prepared by the method described above, after which it was poured into prepared PLA molds in 100 μL and irradiated with UV (365 nm) for 10 min in the chamber of a Rokit Invivo 3D printer. After the end of polymerization, the scaffolds were taken out of the molds, placed in a sterile Petri dish, and filled with DMEM medium (PanEco, Moscow, Russia) to prevent drying. The finished bulk scaffolds (Figure 8B) were 8 mm in diameter and 2.1 mm high. On the same day, scaffolds were implanted subcutaneously in the withers of six outbred white male rats (body weight ~200 g, age 2.5 months). All operations with animals were performed under ether inhalation anesthesia. Hair was removed from the operation site. The operating field was treated with 70% ethanol solution. An incision was made with scissors and a pocket was formed under the skin with a scalpel, into which the implant was placed. When the wound was sutured, the edges of the pocket were tightened, the site of implantation was marked with a colored thread of suture material (Monocryl Poliglecaprone 25, Ethicon). The seam was treated with a 3% hydrogen peroxide solution. For better fixation, medical glue was applied on top. The area around the surgical field was additionally anesthetized with 0.5% novocaine solution. During the experiment, the rats were isolated from each other in single cages with free access to water and food. The surgical site was inspected daily. Two animals were euthanized on the 10th, 17th, and 33rd days after implantation, and the material was taken for histological examination. All work with laboratory animals was in accord with the local commission for bioethical control (No. 1-H-00022 dated 5 July 2022).

### 4.9. Histological Studies

Scaffolds with fragments of surrounding tissues were fixed for 24 h in Bouin’s acidic solution (1.3% trinitrophenol (Sigma-Aldrich, St. Louis, MO, USA), 40% formalin (hereinafter BioVitrum, unless otherwise indicated)). After washing in 70% ethanol, standard histological preparation of samples was performed, after which they were placed in paraffin medium (Histomix, Little Rock, AR, USA). Paraffin sections 5 µm thick were obtained using a microtome (Leica RM2235, Miami, FL, USA) and placed on silanized slides (S3003, Dako). Deparaffinized sections were stained with hematoxylin and eosin (8GX, Sigma-Aldrich, St. Louis, MO, USA). Sections were dehydrated in alcohol, cleaned with (ortho-)xylene, and embedded with Canadian balsam (Merck, Boston, MA, USA). Histological sections were examined under an AXIO Imager A1 microscope (Carl Zeiss, Jena, Germany) with a Canon Power Shot A640 (Canon, Ohta-ku, Tokyo, Japan) camera.

### 4.10. Statistical Analysis

The average values of the indicators and the standard error of the mean were calculated to present the results. The cell count error in the Goryaev chamber was evaluated according to the Poisson distribution. The comparison was carried out using analysis of variance and Tukey’s and Student’s tests. Differences were considered statistically significant at *p* < 0.05. The data was visualized using the Veusz graphical application version 1.23.2.

## 5. Conclusions

Gelatin methacryloyl (GelMA) hydrogels are widely used in tissue engineering because gelatin provides cells with biological signals and its functionalization allows the physicochemical properties to be adapted to the requirements of different tissues. In this work, we synthesized GelMA with a high degree of substitution (82.75 ± 7.09%), and investigated its printability, cyto-, and biocompatibility. A satisfactory printing quality of 15% hydrogel was demonstrated. However, the high degree of substitution combined with a 10% concentration reduced the adhesive properties of the hydrogel and slowed its degradation in animals after subcutaneous implantation.

## Figures and Tables

**Figure 1 ijms-24-02121-f001:**
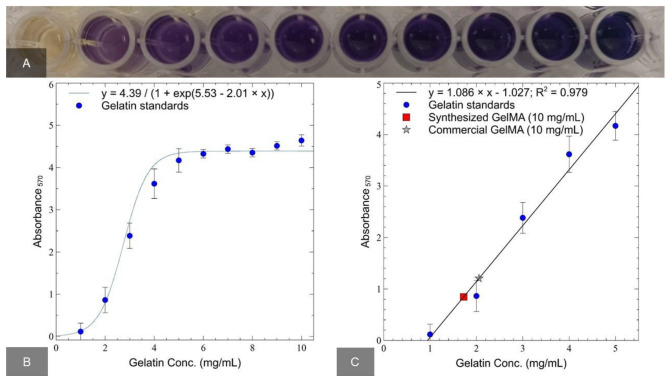
Ninhydrin assay. An example of coloring gelatin at a concentration of 1–10 mg/mL to obtain a standard curve using ninhydrin analysis (**A**). The value of the optical density of gelatin from 1–10 mg/mL (**B**). The linear section of the standard curve, which was used to build a linear approximation (**C**).

**Figure 2 ijms-24-02121-f002:**
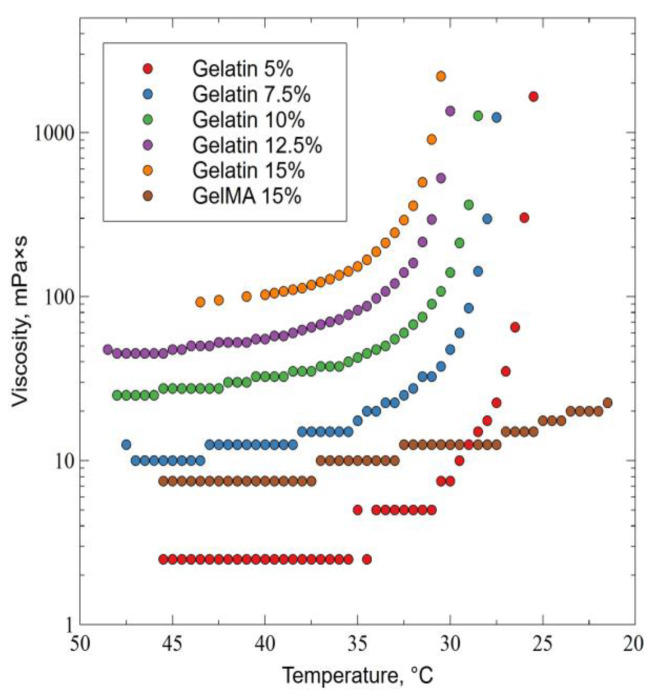
Dependence of the viscosity of hydrogels GelMA 15% and gelatin 5%, 7.5%, 10%, 12.5%, and 15% on temperature.

**Figure 3 ijms-24-02121-f003:**
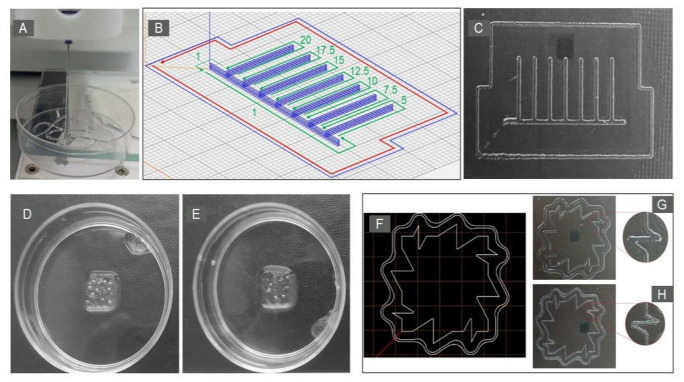
Printability of synthesized hydrogel. Formation of a 15% GelMA filament during extrusion through a 21G needle (**A**). The appearance of the maximum print speed estimation panel (**B**). The blue line marks the printed object, the red line marks the direction of ‘Skirt’ command printing at 1 mm/s, and the green line marks the direction of object printing at the speed range under study. The resulting panel (**C**). An example of an object printed from 15% GelMA before (**D**) and after a 72 h incubation (**E**). The appearance of the angled print panel (**F**) and the resulting panel at a speed of 5 mm/s (**G**) and 1 mm/s (**H**).

**Figure 4 ijms-24-02121-f004:**
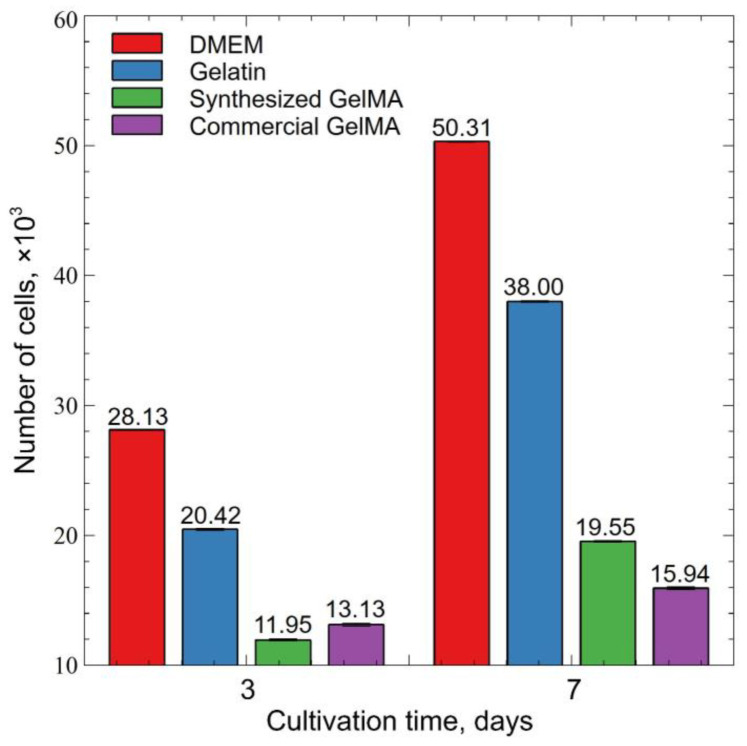
Proliferation of ADSCs in DMEM without gelatin, in the presence of unmodified porcine skin gelatin (type A, Bloom 300), synthesized, and commercialized by GelMA. For each group, the number of cells ($\bar{X}\pm S_{\bar{X}}$) is shown on the 3rd and 7th days after seeding.

**Figure 5 ijms-24-02121-f005:**
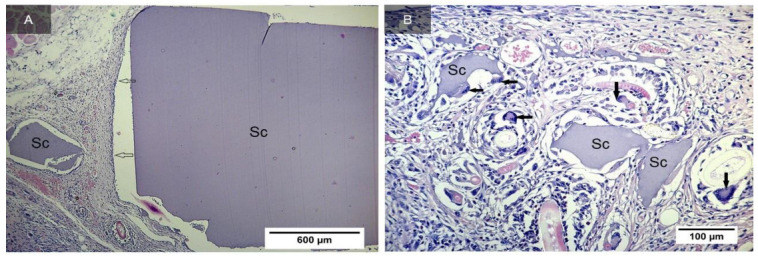
Implant from the synthesized GelMA in the subcutaneous tissue of rats after 10 days (indicated by the letters Sc). The capsule is shown by arrows. Staining with hematoxylin and eosin. Magnification of the image—×5 (**A**). Fragments of an implant in a connective tissue capsule (indicated by the letters Sc) surrounded by multinucleated macrophages (shown by arrows). Staining with hematoxylin and eosin. Magnification of the image—×20 (**B**).

**Figure 6 ijms-24-02121-f006:**
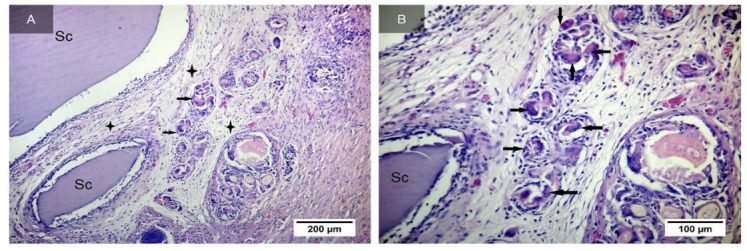
Implant from synthesized GelMA in the subcutaneous tissue of rats after 17 days (indicated by the letters Sc). Connective tissue capsule around the implant (shown with asterisks). Multinucleated macrophages destroying scaffold fragments (shown by arrows). Staining with hematoxylin and eosin. Magnification of the image—×10 (**A**). Enlarged fragment. Arrows show multinucleated scaffold resorption cells. Staining with hematoxylin and eosin. Magnification of the image—×20 (**B**).

**Figure 7 ijms-24-02121-f007:**
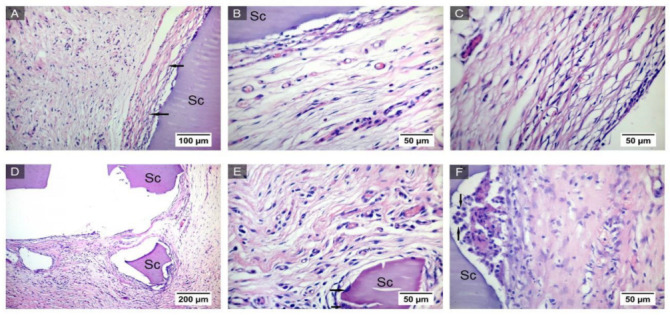
Implant from synthesized GelMA in the subcutaneous tissue of rats after 33 days (marked with Sc), surrounded by a connective tissue capsule. Connective tissue capsule around the implant (shown by arrows). Staining with hematoxylin and eosin. Magnification of the image—×20 (**A**). Collagen fibers in the connective tissue capsule. Staining with hematoxylin and eosin. Magnification of the image—×40 (**B**). Connective tissue of the capsule around the implant. Staining with hematoxylin and eosin. Magnification of the image—×40 (**C**). Fragments of the implant. Staining with hematoxylin and eosin. Magnification of the image—×10 (**D**). Magnification of the image—×40. Arrows show multinucleated macrophages (**E**). Circumcellular infiltrate (marked with arrows) in the border zone between the capsule and the implant (marked with Sc) after 33 days. Hematoxylin and eosin staining. Magnification of the image—×40 (**F**).

**Figure 8 ijms-24-02121-f008:**
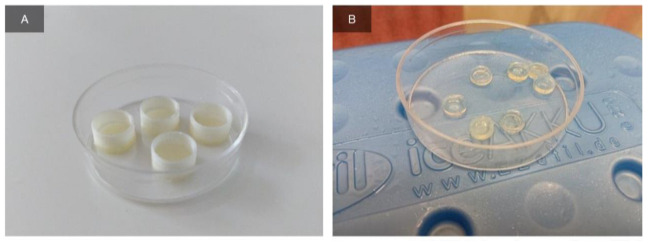
Scaffold preparation for in vivo implantation. Photograph of the printed PLA form (**A**). Scaffolds obtained by molding (**B**).

**Table 1 ijms-24-02121-t001:** Results of measuring the area of objects from 15% GelMA before and after 72 h incubation.

X ± S (CI, 95%, *n* = 3)	
After printing	After incubation
0.721 ± 0.033 (0.639, 0.803)	0.820 ± 0.020 (0.772, 0.869)

## Data Availability

The study did not report any data.

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
