# Peer review of "Properties and Printability of the Synthesized Hydrogel Based on GelMA"

_ijms, 2023, doi:10.3390/ijms24032121_

Round 1

Reviewer 1 Report

In the article: “Properties and Printability of the Synthesized Hydrogel Based on GelMA”, the authors discussed about the properties of GelMA with a high degree of substitution of 82.75 ± 7.09%, its  suitability for extrusion printing, cytocompatibility and biocompatibility.

Overall, this manuscript results very interesting, the authors clearly explain the rational of the study and discussed the topic point by point.

However, we would like to invite the authors  to clarify some minor points:

 1.       Please check the check punctuation and spaces;

2.  Among the introduction, the authors described the properties of biomaterials based on natural or artificial polymers and the modified gelatin to improve its biophysical properties. In this respect the following reference should be useful: “Vassallo V, Tsianaka A, Alessio N, Grübel J, Cammarota M, Tovar GEM, Southan A, Schiraldi C. Evaluation of novel biomaterials for cartilage regeneration based on gelatin methacryloyl interpenetrated with extractive chondroitin sulfate or unsulfated biotechnological chondroitin. J Biomed Mater Res A. 2022 Jun;110(6):1210-1223. doi: 10.1002/jbm.a.37364. Epub 2022 Jan 28. PMID: 35088923; PMCID: PMC9306773” ;

3.  The PBS used for the preparation of gels was with or without calcium and magnesium? Please specify;

4.     Did you evaluated the release of hydrogel from GelMa scaffold?

5.    In which way the cells were counted? Did you not perform a viability assay? MTT? CCK-8? Why?

6.   By histology analyses it is possible to observe the presence of immunitary cells? It is ongoing an inflammatory process?

7.  Did you evaluate the expression/production of specific biomarkers of inflammation?

Reviewer 2 Report

The paper submitted by Arguchinskaya et al. investigates the printability and the biological properties of some hydrogels based on two gelatin methacryloyl samples, one commercial and another synthesized. 

The manuscript is clear, well written and the conclusions are supported by the results. However, some corrections are needed in order to improve the overall quality of the paper:

1. the authors must better highlight which is the originality and the novelty of the study. they must discuss in detail about the already published papers on this subject.

2. the viscosity at 00C must be determined because at this temperature the printability process occurs.

3. verify the abscissa axis in fig 2.

4. add additional details about the type of the lamp used for photo-crosslinking

5. swelling during 33 days must be carried out as it is important to determine if the hydrogel scaffolds are still intact after this period.

Round 2

Reviewer 2 Report

The paper can be published as it is.